# Institution and gender-related differences in publication speed before and during COVID-19

**Claudia Acciai**[2]*, **Benjamin C. Holding**[1,2]*, **Jesper W. Schneider**[3], **Mathias W. Nielsen**[2]

**1** Department of Clinical Neuroscience, Karolinska Institutet, Stockholm, Sweden, **2** Division of Sociology, University of Copenhagen, Copenhagen, Denmark, **3** Department of Political Science, Danish Centre for Studies in Research and Research Policy, Aarhus University, Aarhus C, Denmark

* cla@soc.ku.dk (CA); benjamin.holding@ki.se (BCH)

**Editor:** Negar Rezaei, Non-Communicable Diseases Research Center, Endocrinology and Metabolism Population Sciences Institute, Tehran University of Medical Sciences, ISLAMIC REPUBLIC OF IRAN

**Data Availability Statement:** Data are available on the Open Science Framework (https://osf.io/zebka/).

## Abstract

The COVID-19 pandemic elicited a substantial hike in journal submissions and a global push to get medical evidence quickly through the review process. Editorial decisions and peer-assessments were made under intensified time constraints, which may have amplified social disparities in the outcomes of peer-reviewing, especially for COVID-19 related research. This study quantifies the differential impact of the pandemic on the duration of the peer-review process for women and men and for scientists at different strata of the institutional-prestige hierarchy. Using mixed-effects regression models with observations clustered at the journal level, we analysed newly available data on the submission and acceptance dates of 78,085 medical research articles published in 2019 and 2020. We found that institution-related disparities in the average time from manuscript submission to acceptance increased marginally in 2020, although half of the observed change was driven by speedy reviews of COVID-19 research. For COVID-19 papers, we found more substantial institution-related disparities in review times in favour of authors from highly-ranked institutions. Descriptive survival plots also indicated that scientists with prestigious affiliations benefitted more from fast-track peer reviewing than did colleagues from less reputed institutions. This difference was more pronounced for journals with a single-blind review procedure compared to journals with a double-blind review procedure. Gender-related changes in the duration of the peer-review process were small and inconsistent, although we observed a minor difference in the average review time of COVID-19 papers first authored by women and men.

## 1. Introduction

The COVID-19 pandemic has exacerbated inequalities in science. Gender gaps in authorships, publication outputs, submission rates and invited journal contributions have widened [1–11], and young scientists, especially those with children, have seen substantial decreases in their weekly time for research [12, 13]. Studies also indicate intensified global disparities in

**Funding:** This study was funded by Carlsbergfondet (the Carlsberg foundation) – Award # CF19-0566. P.I. M.W.N] The funders had no role in study design, data collection and analysis, decision to publish, or preparation of the manuscript.

**Competing interests:** The authors have declared that no competing interests exist.

publication outputs, with a decline in articles from developing countries [14, 15] and with few developing countries directly involved in COVID-19 related research [16].

The scientific community reacted quickly to the pandemic. Journal submissions surged [17–19], and editors and reviewers were fast to respond to the need for evidence on the prevention, diagnosis and treatment of the virus [14, 20]. However, diligent peer reviewing is time-demanding, and in the wake of the pandemic, the average review time–from submission to publication–decreased by 49% according to an analysis of articles published in 14 medical journals, prior to and during COVID-19 [21]. This decrease was primarily driven by speedy reviews of COVID-19 related submissions, while no acceleration was observed for manuscripts on other topics [22, 23].

The disruption of the publication system imposed by COVID-19 may have led to gender and institution-related disparities in peer reviewing for two reasons. First, editorial decisions during the pandemic were made under intensified time constraints with a substantial hike in journal-submissions and a global push to get COVID-19 research quickly through the review process. Research suggests that such time constraints can exacerbate status biases in complex decision-making processes [24–28], and both gender and institutional affiliation have been highlighted as "status signals" that may implicitly influence evaluative judgments in favour of male researchers and scientists from high status institutions [29–35]. Second, evidence suggests that women's working conditions have been disproportionately affected by the pandemic due to disparities in teaching, service and caregiving loads [13, 36–40]. The increase in time constraints is likely reflected in the publication speed of women scientists, who, on average, may have been slower to respond to resubmission requests, hence delaying the duration of the peer-review process. Similarly, also institution-related disparities may have been exacerbated by the pandemic. Scientists at less affluent research institutions may have lacked the necessary resources, infrastructure and support to adapt quickly to the changing working conditions imposed by COVID-19, which may also have prolonged the review process.

In this study, we adopt a new perspective on the widening disparities during COVID-19 by using recently available data on the peer-review duration of medical papers. We aim to examine whether the pandemic has widened gender and institution-related differences in the average time from manuscript submission to acceptance. Additionally, given that some peer review processes differ in terms of whether institutional status queues and/or gender are visible to reviewers (due to having either single or double blinded review), we also descriptively investigate whether double blind peer review reduces institution/gender biases in publication speed during the pandemic.

## 2. Materials and methods

On January 11, 2021, we searched PubMed Medline for papers including COVID-19 related keywords ('COVID-19', 'COVID', 'SARS-CoV-2', 'severe acute respiratory syndrome coronavirus 2', '2019-nCoV', '2019 novel coronavirus', 'Wuhan coronavirus') [41] in their titles or abstracts. These articles needed to be published between March 11, 2020 (the date when the World Health Organization declared COVID-19 a global pandemic) and December 31, 2020. This query returned 106,116 articles (S1 Fig in S1 Appendix). After removing the articles with missing information about submission-acceptance dates (N = 64,530), we restricted our focus to outlets that had published > 50 articles with COVID-19 related keywords resulting in a core sample of 80 medical and health-science journals and a final sample of 8,828 COVID-19 related research articles. Next, we ran queries to establish two control samples: one consisting of articles *without* COVID-19 related keywords published in the same 80 journals during the pandemic (March 11, 2020, to December 31, 2020, N = 82,226) and another consisting of

articles *without* COVID-19 related keywords published in the same journals during 2019 (March 11, 2019, to December 31, 2019, N = 64,685). In these datasets, we also removed entries with missing information. In total, our final sample consists of 78,085 papers distributed across the following three subsamples: 2020 COVID-19 related papers (N = 8,828), 2020 non-COVID-19 related papers (N = 40,125) and 2019 control papers (N = 29,132).

Data on the full names and affiliations of first- and last authors were obtained by matching the 78,085 PubMed records to article meta-data from a structured version of Web of Science hosted at the Centre for Science and Technology Studies (CWTS), Leiden University. The CWTS version of the WoS database also provides an estimated gender for individual researchers based on three name-to-gender assignment algorithms (Gender API, Gender Guesser and Genderize.io). This approach estimates a given author's likelihood of being a man or a woman based on full names and country affiliations, and has been shown to be ~96% accurate [42]. We were able to use this database to assign an estimated gender of 90% of the single authors, 71% of the first authors and 82% of the last authors.

We used information from the Leiden Ranking (2019) and the QS World University Ranking (2019) to infer the reputation of each first and last author's institutional affiliation. From the former, we relied on the PP top-10% indicator, which ranks universities based on their proportion of publications belonging to the top 10% most cited globally. PP top-10% is a yearly measure calculated based on field-normalized citations [43]. From the latter, we adopted the QS comprehensive ranking (averaged across the last three years), which provides a proxy of a university's worldwide reputation, with 40% of the underlying ranking data derived from a global reputation survey. Of the 73,902 articles with eligible metadata in WoS, 48,270 had either first or last authors with institutional affiliations covered by the Leiden Ranking, while 30,900 had either first or last authors with affiliations covered by the QS-ranking.

## 2.1 Outcome measure

Our outcome variable is a count measure of the number of days from the initial submission of a manuscript to its acceptance for publication (i.e., *peer-review duration*). Information on the submission and acceptance dates of each article was retrieved from PubMed Medline.

## 2.2 Main predictors

*Gender* was computed as a three-factor variable (man, woman, unknown gender). This third gender category covers all the authors for which the name-to-gender assignment algorithms did not provide reliable results. *Institution status (Leiden)* and *Institution status (QS)* are continuous variables measuring the status of the first and last authors' institutional affiliations according to the PP-top-10% and QS-rankings. These variables have been re-scaled to obtain standardized measures centred on their overall mean. Following Gelman [44], we divided each variable by two standard deviations to allow the numeric inputs to be interpreted on the same scale as the binary case variables (i.e. year and COVID-19). A one unit change on the rescaled measure of institution status should be interpreted as a two-standard deviation change on a scale from a lower ranking to a higher ranking institution.

## 2.3 Covariates

*Year* is a binary variable that measures whether a research article was published in 2019 or 2020 (2019 = 0, 2020 = 1). We interacted Year with the Gender and Institution Status in Models 1 to 4, since we were interested in changes occurring in the wake of the pandemic. COVID-19 was computed as a binary variable that measures whether a research article includes COVID-19 related keywords in its title/abstract or not (No COVID-19 keywords = 0, COVID-

19 keywords = 1). COVID-19 was used as a covariate in Models 2 and 4, and interacted with Gender and Institution Status in Models 5 and 6. Journal Impact Factor is a continuous variable that measures the average number of citations of articles published in each medical journal within the preceding two years. This variable was used as a covariate in all models to adjust for variations in the prestige of the various outlets, and has been rescaled by dividing by two standard deviations [44]. *Peer-review Procedure* is a dichotomous measure that specifies if a journal operates with a single-blind or a double-blind review (double-blind = 1, single-blind = 0) procedure. In the double-blind procedure, the identity of the author is kept hidden from the reviewers, while, in the single-blind condition, both editors and reviewers know the identity of the author. Information on peer-review procedure was obtained by screening journal websites and by contacting journal editors by email. Descriptive statistics for all variables can be found in S1 Table in S1 Appendix. To measure the relationship between our main predictors (Institution Status and Gender) and the number of days from manuscript submission to acceptance (peer-review duration) we used linear-mixed effects models [45], with papers nested in journals focusing on first author (Model 1; 2; 5) and last author (Model 3; 4; 6) information. This approach allowed us to account for journal-level variation in average review times [46]. To examine the sensitivity of the regression results to alternative model specifications, we carried out robustness checks using Poisson generalized mixed models–appropriate for right-skewed count outcomes like ours [47]–and linear regression models with cluster-robust standard errors at the journal level.

We used descriptive survival plots to examine the within-group variability in the publication speed of articles published by authors at highly ranked and lower-ranked institutions. Specifically, we used the Kaplan-Meier non-parametric estimator [48]. This approach visualizes the expected duration of the peer-review process (in days and across groups) up to the point where a paper is accepted for publication. In the Kaplan-Meier survival analysis, we created dummies for authors affiliated with top-ranked and lower-ranked institutions. Authors affiliated with top-ranked institutions are those whose affiliation fall within the $10^{th}$ decile of the Leiden and QS ranking indicators. Lower-ranked universities refer to institutions that fall within the $5^{th}$ decile or lower on the same measures. We also included peer-review procedure as a factor in the survival plots.

The statistical analyses were conducted in R version 4.0.2. For the linear mixed effects models and the Poisson generalized mixed model, we used the 'lme4' 1.1–27.1 package [49], and for the linear regression models with cluster-robust standard errors, we used the 'estimatr' v. 0.30.2 package [50]. We used 'Sjplot' v. 2.8.9 package [51] to produce Fig 1, and the 'survminer' v. 0.4.9 package [52] to produce Figs 2 and 3.

## 3. Results

### 3.1. Institution status

Our analysis suggests that institution-related disparities in the average time from manuscript submission to acceptance have slightly increased during the pandemic. As indicated by the interaction between Institution Status and Year in Table 1 (Models 1 and 3), the marginal difference between the publication speed of papers from higher ranked and lower ranked universities (measured by two standard deviations difference on the Leiden Ranking) increased by 1.98 days (99% CI: -3.48 to -0.48) for first authors and by 2.15 days (99% CI: -3.71 to—0.6) for last authors in 2020 compared to 2019. Comparing review times for top-and bottom-ranked institutions that are two standard deviations below and above the mean of the ranking distribution, this corresponds to a marginal difference of ∼ 3.96 days for first authors and ∼ 4.3 days for last authors (S2 Fig in S1 Appendix). Given that the average duration of the peer-

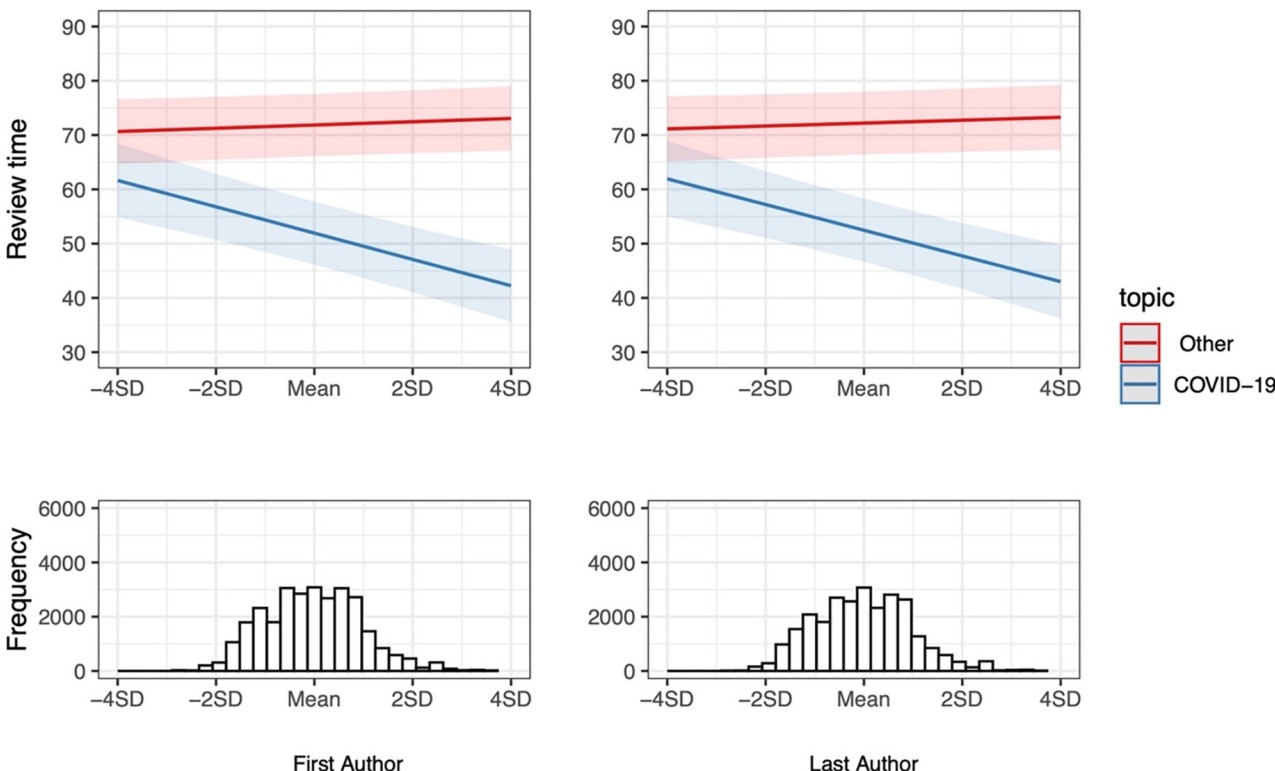

**Fig 1. Estimated difference in average review time of COVID-19 papers (red line) and other medical research (blue line) published in 2020 by authors from lower and higher ranked institutions.** The Y-axis reports the number of days from manuscript submission to acceptance. The X-axis scores Institution status (according to the Leiden Ranking's PP-top 10% measure) by standard deviations. The light red and light blue areas around the lines indicate the 99% confidence intervals. The histograms below the interaction plots display the frequency distribution of papers across various levels of Institution status.

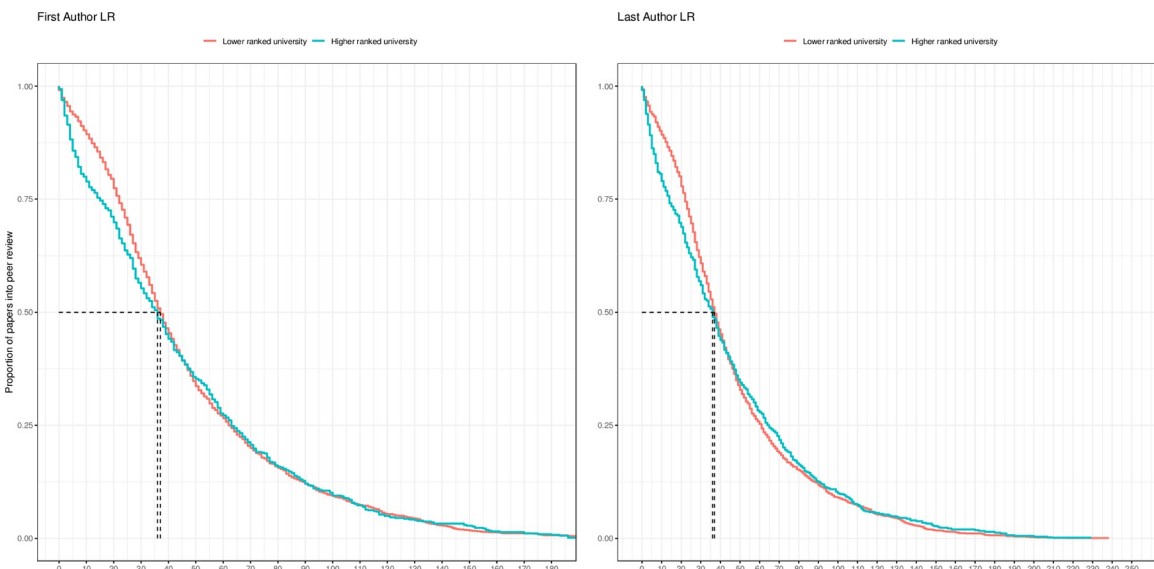

**Fig 2. Survival plots of time from submission to publication for COVID-19 papers authored at higher-ranked (teal line) *versus* lower ranked universities (red line).** The groups of highly ranked and lower ranked universities are inferred from the Leiden Ranking. Dashed black lines specify the difference in median review times of papers from highly ranked and lower ranked institutions.

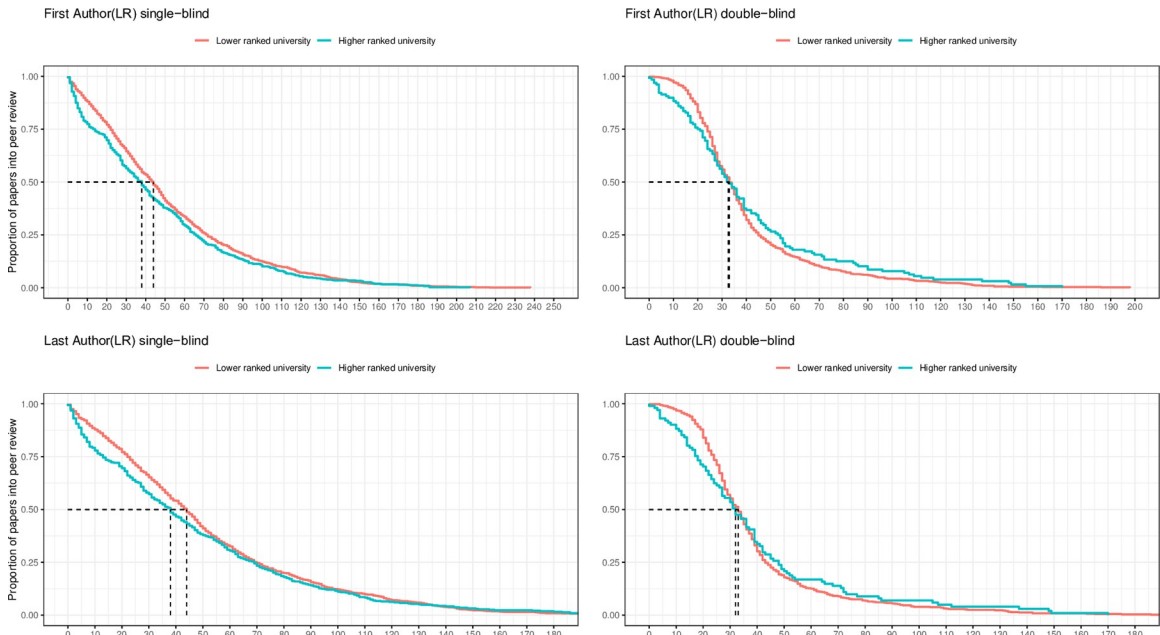

**Fig 3. Survival plots of time from submission to publication for COVID-19 papers published in single-blind *versus* double-blind peer reviewed journals, authored at higher-ranked (teal line) *versus* lower ranked universities (red line).** The groups of highly ranked and lower ranked universities are inferred from the Leiden Ranking. Dashed black lines specify the difference in median review times of papers from highly ranked and lower ranked universities.

**Table 1. Mixed linear regression models predicting the change in peer-review duration from 2019 to 2020.**

**Linear mixed effect models**

| Predictors | M1 peer-review time (First authors) Estimates | M1 99% CI | M2 peer-review time (First authors) Estimates | M2 99%CI | M3 peer-review time (Last authors) Estimates | M3 99%CI | M4 peer-review time (Last authors) Estimates | M4 99%CI |
|---|---|---|---|---|---|---|---|---|
| Intercept | 76.34 | 68.90 – 83.78 | 79.77 | 72.41 – 87.12 | 76.45 | 68.94 – 83.96 | 79.94 | 72.54 – 87.33 |
| Institution Status (LR) | 0.61 | -0.58 – 1.81 | 0.12 | -1.06 – 1.29 | 0.82 | -0.41 – 2.05 | 0.37 | -0.84 – 1.58 |
| Year | -6.78 | -7.98 – -5.59 | -4.09 | -5.28 – -2.91 | -6.33 | -7.78 – -4.88 | -3.74 | -5.18 – -2.31 |
| Journal Impact Factor | -0.66 | -6.81 – 5.49 | -0.53 | -6.61 – 5.55 | -1.52 | -7.74 – 4.70 | -1.24 | -7.36 – 4.88 |
| Gender (male) | -0.87 | -2.20 – 0.45 | -1.05 | -2.35 – 0.25 | -0.54 | -1.94 – 0.86 | -0.60 | -1.98 – 0.78 |
| Gender (unknown) | 0.71 | -0.82 – 2.25 | 0.96 | -0.55 – 2.47 | 1.47 | -0.30 – 3.24 | 1.77 | 0.03 – 3.51 |
| Institution Status (LR)*Year | -1.98 | -3.48 – -0.48 | -0.83 | -2.30 – 0.65 | -2.15 | -3.71 – -0.60 | -1.10 | -2.64 – 0.43 |
| Year*Gender (male) | -1.02 | -2.71 – 0.67 | 0.23 | -1.43 – 1.88 | -1.07 | -2.86 – 0.73 | -0.37 | -2.14 – 1.39 |
| Year*Gender (unknown) | 0.45 | -1.55 – 2.45 | 0.46 | -1.51 – 2.42 | 0.12 | -2.18 – 2.42 | 0.22 | -2.04 – 2.48 |
| COVID-19 | | | -22.55 | -23.96 – -21.14 | | | -22.51 | -24.01 – -21.02 |
| **Random Effects** | | | | | | | | |
| $\sigma^2$ | 966.70 | | 934.02 | | 954.03 | | 922.91 | |
| $\tau_{00}$ | 623 journal | | 605.53 journal | | 626.27 journal | | 606.54 journal | |
| ICC | 0.39 | | 0.39 | | 0.40 | | 0.40 | |
| N | 80 journals | | 80 journals | | 80 journals | | 80 journals | |
| Observations | 48269 | | 48269 | | 44571 | | 44571 | |
| Marginal $R^2$ / Conditional $R^2$ | 0.008 / 0.397 | | 0.034 / 0.415 | | 0.008 / 0.403 | | 0.033 / 0.418 | |
| AIC | 469188.389 | | 467530.623 | | 432676.812 | | 431199.268 | |

**Table 2. Mixed linear regression models predicting the moderating effect of the COVID-19 topic on peer-review duration.**

**Linear mixed effect models**

| Predictors | M5 peer-review time (First authors) | | M6 peer-review time (Last authors) | |
|---|---|---|---|---|
| | Estimates | 99%CI | Estimates | 99%CI |
| Intercept | 71.62 | 64.09–79.14 | 72.06 | 64.50 – 79.61 |
| Institution Status (LR) | 0.60 | -0.40 – 1.61 | 0.55 | -0.49 – 1.59 |
| COVID-19 | -18.01 | -20.25 – -15.78 | -18.60 | -21.32 – -15.88 |
| Journal Impact Factor | -0.79 | -7.05 – 5.46 | -1.55 | -7.87 – 4.77 |
| Gender (male) | -0.27 | -1.34 – 0.81 | -0.47 | -1.61 – 0.66 |
| Gender (unknown) | 1.83 | 0.50 – 3.15 | 2.00 | 0.46 – 3.53 |
| Institution Status* COVID-19 | -5.49 | -7.94 – -3.05 | -5.43 | -8.03 – -2.83 |
| COVID-19 *Gender (male) | -3.20 | -5.97 – -0.43 | -2.22 | -5.35 – 0.90 |
| COVID-19 *Gender (unknown) | -1.97 | -5.53 – 1.59 | 1.02 | -3.08 – 5.11 |
| **Random Effects** | | | | |
| $\sigma^2$ | 840.97 | | 833.47 | |
| $\tau_{00}$ | 636.20 journal | | 636.78 journal | |
| ICC | 0.43 | | 0.43 | |
| N | 80 journals | | 80 journals | |
| Observations | 28970 | | 26686 | |
| Marginal $R^2$ / Conditional $R^2$ | 0.037 / 0.452 | | 0.035 / 0.453 | |
| AIC | 277689.924 | | 255576.412 | |

review process was ∼70 days for first authors and last authors in 2020, these marginal differences seem relatively small. The observed change appears to be partially explained by speedy reviews of COVID-19 research. Indeed, when COVID-19 is factored into the analysis (Table 1, Models 2 and 4), the coefficients for the interaction decrease to -0.83 days (99% CI: -2.30 to 0.65) for first authors and to -1.10 days (99% CI: -2.64 to 0.43) for last authors.

We obtain comparable results when Institution Status is calculated based on the QS ranking as opposed to the Leiden Ranking (S2 Table in S1 Appendix). Robustness checks based on Poisson generalized mixed models and linear regression models with cluster-robust standard errors at the journal level yield qualitatively similar results (S4 and S5 Tables in S1 Appendix).

Zeroing in on papers published in 2020 (Table 2), we observe an 18.01 days (99% CI: -20.25 to– 15.78), and 18.6 days (99% CI: -21.34 to -15.88) shorter average review time for first and last authored papers from higher ranked universities that focus on COVID-19 compared to papers on other topics (Models 5 and 6). As indicated by the interaction between Institution Status (measured by the Leiden ranking) and COVID-19, two standard deviations move-up in the institutional prestige hierarchy reduces this average review time with an additional 5.49 days (99% CI: -7.94 to -3.05) for first author and 5.43 days (99% CI: -8.03 to -2.83) for last authors.

Fig 1 plots this interaction and indicates that the review time of papers on other topics than COVID-19 (red line) does not vary substantively by institution status. However, in the case of COVID-19 papers (blue line), the status variation is obvious. Comparing universities that are two standard deviations below and above the mean of our measure of institution status, we find that COVID-19 papers with first authors and last authors at highly-ranked universities see an 18–20% shorter average review time than COVID-19 papers with first and last authors at low-ranked universities.

These marginal differences are slightly more pronounced when institution status is calculated based on the QS ranking, with an estimated gain of 5.8 days (99% CI: -8.65- -2.96) for

first authors and -6.58 days (99% CI: -9.75- -3.41) for last authors for every two standard deviations increase in institution status (S3 Table in S1 Appendix). Robustness checks based on alternative model specifications yield comparable results (S4 and S5 Tables in S1 Appendix).

## 3.2. Gender

Our analysis of gender-related changes in the duration of the peer-review process indicates small and inconsistent effects. As demonstrated by the interaction between Gender and Year in Table 1 (Models 1 and 3), the change in the average gender difference in review time from 2019 and 2020 is negligible (first authors, marginal gender difference = -1.02 days, 95% CI: -2.71 to 0.67; last authors, marginal gender difference = 1.07 days, 95% CI: -2.86 to 0.73). However, according to the interaction between Gender and COVID-19 in the subsample of papers published in 2020 (Table 2, Model 5 and 6), the average review time of COVID-19 papers by men first authors is 3.20 days (99% CI: -5.97 to -0.43) shorter than the average review time of COVID-19 papers by women first authors. This marginal gender difference is less pronounced in the subsample of last authors, with an estimated reduction of 2.22 days for men (99% CI: -5.35 to 0.90) (Table 2, Model 6). Robustness checks indicate comparable results across alternative model specifications (S4 and S5 Tables in S1 Appendix).

## 3.3. Survival plots

Further analyses indicate that scientists with prestigious affiliations have benefitted slightly more from fast-track peer reviewing during COVID-19 than colleagues from less reputed institutions. In Fig 2, we used survival plots to visualize within-group variability in the publication speed of COVID-19 articles by authors at highly ranked and lower-ranked institutions in 2020. The X-axes in the plots denote the length of the peer-review process in number of days, while the Y-axes show the proportion of papers still under review over the total number of submitted papers (e.g., 0.75 means that 75% of papers are still under review while 25% have been published).

As shown in the figure, we observe a slightly faster decline in the proportion of papers from highly-ranked institutions (teal line) on the left side of the X-axis, compared to papers from lower-ranked institutions (red line). Indeed, while ∼25% of papers from highly ranked institutions were published within 15 days from submission, this was only the case for ∼16–17% of the papers from lower ranked institutions. Trends are similar when the distinction between highly-ranked and lower-ranked institutions is based on the QS-ranking as opposed to the Leiden ranking (see S5 Fig in S1 Appendix).

In Fig 3, we explore how the publication speed of COVID-19 papers varies by peer-review procedure. Specifically, we compare the trajectories of papers from highly ranked and lower ranked institutions that have been submitted to journals with single- or double-blind peer-review procedures. As shown in the upper left and right panels, the trajectories of first-authored papers from highly ranked and lower ranked institutions (measured by the Leiden ranking) differ slightly across the single-blind and double-blind review procedures. The median difference in peer-review between higher and lower ranked institutions is ∼7 days for papers in single-blind journals (left panel) and ∼1 day for papers in double-blind journals (right panel). As shown in the left panel, the trend line for papers from highly ranked institutions shows a steep decline from day 0 to day 10 and is below the trend line for papers from lower-ranked institutions until day 110. In comparison, the trend lines for papers from highly ranked and lower ranked institutions undergoing double-blind review intersect at day 32–33. Results for the sample of last authors (lower panels) are comparable. In the complementary analysis, where top-ranked and lower-ranked universities are identified based on the QS-

ranking, the advantage for top-ranked universities under the single-blind procedure compared to the double-blind procedure is slightly more pronounced (S6 Fig in S1 Appendix).

As a counterfactual analysis, we ran the same survival analysis as presented in Fig 3 but based on the 2019 and 2020 samples of papers on other topics than COVID-19. In these comparisons status-related differences in the publication speed across double blind and single-blind journals were miniscule (S8-S11 Figs in S1 Appendix).

## 4. Discussion

The current study aimed to assess whether the pandemic had widened existing institution and gender-related differences in publishing speed, by using bibliometric metadata on the duration of peer-review. While prior studies have used such data to document increases in the average publication speed during COVID-19 [17–19, 53], none have examined how changes in journal turn-around times varied by gender and institution status.

Using paper-level data from 80 medical journals, we show that researchers at highly ranked institutions have increased their publication speed slightly more in 2020 than have scientists at less reputed institutions. However, the average difference is small, and should be interpreted with caution (more on this below). We observe a more substantial status-related difference for COVID-19 research (compared to research on other topics), with papers from top-ranked institutions seeing faster review times than papers from lower-ranked institutions. Moreover, survival plots indicate that scientists with prestigious affiliations have benefitted the most from fast-track peer reviewing and especially so in journals with single-blind review procedures. Finally, our analysis of gender-related changes in publishing speed indicates small and inconsistent effects although we observe a slight difference in the average review time of COVID-19 papers first authored by women and men.

Consistent with previous evidence on COVID-19 [3, 14–20] and earlier health emergencies [54, 55], our analysis shows that scientists (and editors) have responded quickly to the pandemic by speeding up the publication process. On average, we observe a 23 day shorter review time for COVID-19 research compared to papers on other topics (Table 1). The need for quick information to support policy makers and practitioners in making evidence-based choices [18], and the resulting increase in publication speed may have accentuated status biases in peer-reviewing by advantaging researchers at prestigious locations.

At the same time, the observed institution-related disparities may be driven by (average) differences in the quality and immediate relevance of COVID-19 papers coming out of more and less affluent institutions. As mentioned earlier, scientists at less affluent research institutions may have lacked the necessary resources, infrastructure and support to adapt quickly to the changing working conditions imposed by COVID-19. This may have lowered the average quality of their submissions, while also making them slower to respond to resubmission requests, hence delaying the duration of the peer-review process.

While the present study does not allow us to disentangle the influence of these proposed mechanisms, it highlights a previously overlooked disparity in peer-reviewing during COVID-19 that has enabled research from high-status institutions to enter more quickly into the public domain.

As for gender, our finding that especially women first-authors (who on average tend to be earlier in their career than women last authors) of COVID-19 papers had a slightly slower publishing speed, align with previous research on gender disparities during the pandemic. According to this literature, lockdowns of schools, universities and daycares took the greatest toll on early career women scientists by considerably reducing their weekly time for research [32–37]. Further, evidence suggests that initial funding for COVID-19 related research was biased

toward male applicants [36] and that women were less likely than men to first-author solicited COVID-19 papers [10]. Hence, lower access to resources and fewer (informal) connections to journal editors may also have slowed down early-career women's publishing speed compared to men's.

Our study has some limitations. First, we lack information on the duration of the peer-review process for rejected papers, which introduces a possible selection bias. If institution and gender-related disparities in journal rejection-rates have widened during COVID-19, this may bias our estimations of disparities in review times downwards.

Second, given the sensitivity of our analysis to uncertainties in the models and data, the small increase in institution- and gender related disparities for papers published in 2020 compared to 2019 should be interpreted with caution. Such uncertainties include database-related errors in the categorization and registration of medical research articles (e.g., spelling mistakes in author names and errors in institutional affiliations in PubMed and Web of Science) and inherent problems associated with non-random sampling procedures [56].

Third, by restricting our analysis to papers from institutions in the Leiden and QS university rankings, we reduced our initial sample by 34% and 58%, respectively. While many of the papers not covered in our analysis likely come out of private organizations, hospitals and health centers, some will be from less affluent research institutions of relevance to our study.

More generally, there are some areas that research may benefit from expanding upon. For example, while the current study accounted for variations at the journal level through a nested design, we did not directly examine how journal characteristics such as open-access status or print vs. online publication models may have influenced the institutional and gender-related differences observed in our study.

Similarly, lack of individual-level researcher data constrained what was possible in the current paper, but future research may benefit from running matched studies (e.g., using propensity-score matching) or in the optimal case within-subjects studies. This would help to reduce implicit differences between groups of interest.

Finally, COVID-19 related changes in editorial workloads, practices and decision criteria that we have not been able to measure may have impacted our findings [18]. Similarly, differences in scientists' willingness and time to take on extra reviewer responsibilities in 2020 may have skewed the reviewer composition towards male evaluators and evaluators from more affluent countries and research institutions [6]. In the future, studies could consider how such changes may have affected journal turn-around times for different author groups.

In summary, our study indicates notable institution-related disparities in the peer-review duration of COVID-19 research compared to medical research on other topics. Future studies should examine how this disparity has affected the available evidence pool on variations in the transmission, prevention and treatment of the Corona virus across geographical regions.

## Supporting information

**S1 Appendix.**
(DOCX)

## Author Contributions

**Conceptualization:** Claudia Acciai, Mathias W. Nielsen.

**Data curation:** Claudia Acciai.

**Formal analysis:** Claudia Acciai, Mathias W. Nielsen.

**Funding acquisition:** Mathias W. Nielsen.

**Software:** Claudia Acciai, Benjamin C. Holding, Jesper W. Schneider.

**Visualization:** Claudia Acciai.

**Writing – original draft:** Claudia Acciai, Mathias W. Nielsen.

**Writing – review & editing:** Benjamin C. Holding, Jesper W. Schneider.

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
