## [Decision Letter · Decision Letter 0]

11 May 2022

PONE-D-22-07162Publishing during the pandemic: Institution and gender-related differences in publication speed before and during COVID-19PLOS ONE

Dear Dr. Holding,

Thank you for submitting your manuscript to PLOS ONE. After careful consideration, we feel that it has merit but does not fully meet PLOS ONE’s publication criteria as it currently stands. Therefore, we invite you to submit a revised version of the manuscript that addresses the points raised during the review process. Please submit your revised manuscript by Jun 25 2022 11:59PM. If you will need more time than this to complete your revisions, please reply to this message or contact the journal office at plosone@plos.org. Please include the following items when submitting your revised manuscript:A rebuttal letter that responds to each point raised by the academic editor and reviewer(s). You should upload this letter as a separate file labeled 'Response to Reviewers'.A marked-up copy of your manuscript that highlights changes made to the original version. You should upload this as a separate file labeled 'Revised Manuscript with Track Changes'.An unmarked version of your revised paper without tracked changes. You should upload this as a separate file labeled 'Manuscript'.

We look forward to receiving your revised manuscript.

Kind regards,

Negar Rezaei, M.D., Ph.D.,

Academic Editor

PLOS ONE

Journal Requirements:

"This study was funded by Carlsbergfondet (the Carlsberg foundation) – Award # CF19-0566.

P.I. M.W.N"

Reviewers' comments:

Reviewer's Responses to Questions

**Comments to the Author**

1. Is the manuscript technically sound, and do the data support the conclusions?

Reviewer #1: Yes

Reviewer #2: Yes

Reviewer #3: Yes

Reviewer #4: Yes

2. Has the statistical analysis been performed appropriately and rigorously? 

Reviewer #1: Yes

Reviewer #2: Yes

Reviewer #3: Yes

Reviewer #4: Yes

3. Have the authors made all data underlying the findings in their manuscript fully available?

Reviewer #1: Yes

Reviewer #2: Yes

Reviewer #3: Yes

Reviewer #4: Yes

4. Is the manuscript presented in an intelligible fashion and written in standard English?

Reviewer #1: Yes

Reviewer #2: Yes

Reviewer #3: Yes

Reviewer #4: Yes

5. Review Comments to the Author

Reviewer #1: Overall:

The manuscript is about disparities in COVID-19 scientific publication. Although the methodology and statistical analysis was robust and solid, there is a considerable room for improvement in the discussion section.

Introduction:

1. While the general storyline of the Introduction is firm and great, I recommend to transfer the exact aim of the study to last paragraph rather than current first paragraph.

2. It is suggested to enrich references 11/12 with a comprehensive study on COVID-19 bibliometrics analysis of countries with regard to their income classification: https://doi.org/10.1371/journal.pone.0258064

Materials and methods:

1. The numbers on retrieved, excluded, and included papers are kind of a result. Please transfer them to the first paragraph of the results section and make sure there is no overlap between sections.

2. Please kindly provide the reference of mentioned reliability statistics fir name-to-gender assignment algorithms.

3. In paragraph on covariates, having COVID-19 keyword or not is more of a binary variable rather than dummy.

Results/Figures/Tables:

1. Please increase the text size in figures, including axis title, legends, and scales.

Discussion:

1. Forth paragraph of the discussion section: The urgent need for rapid research sharing and information dissemination might be generalized to other pandemics as well. It is recommended to broaden your study implications with emphasizing on this generalization and even escalation. For example it is shown in https://doi.org/10.1186/s12889-020-10116-6 that this publication avalanche with probably biased peer-review process is worsening through the time (COVID-19 vs SARS vs Ebola).

Reviewer #2: This is an interesting study and well written. The methodology was well written and covered all details needed for reproducibility and replicability

Kindly find below some minor comments:

In the Abstract, at the results section, “substantial institution-related disparities in review times in favour of authors from highly-ranked institution”. Kindly add “s” to the “institution”.

Using CADTH for the first time, I think it will be appropriate to write it in full like “(Canadian Agency for Drugs and Technologies in Health (CADTH), 2020)”.

Be consistent with your citation format. On page 3 and other sections, you used (author, year), e.g. (Cai et al. 2021)- whiles the general format was Vancouver.

The introduction should end by stating the aim/ objectives of the study. This seems missing in your current manuscript.

First statement in the methodology that is “On January 11, 2021 ……………….and December 31, 2020” needs to be revised to improve readability.

Page 5: “Year will be used as an interaction…”, this should be in the past tense.

At result section, under survival plots (3.3) further analyses indicate that …………………COVID-19 “than” have colleagues from ……..

Kindly delete “than” …..

Same at the above section, in the second statement, “We use survival plots to” should read as “We used survival”.

The authors failed to consider the h-indexes of the first and last authors which could affect the findings of the study. Hence, this should be considered one of the limitations of the study.

The discussion should also highlight on how the quality of papers influence the duration of review. Could this be part of the reasons why articles from institutions of higher repute get published faster?

Reviewer #3: Thank you for submitting your article “Publishing during the pandemic: institution and gender-related difference in publication speed before and during COVID-19”, in which you analysed information about the duration of time between publication and submission dates across 80 journals publishing on topics both COVID and non-COVID during 2019 and 2020, inclusive. You should be applauded on your efforts to collate this amount of data, and analyse it using appropriate statistical approaches.

I have reviewed your paper and provided some suggestions based on the article sections.

Introduction

1. You provided the context of the research nicely here, and your aims might be setter placed at the end of the introduction rather than in the first paragraph

2. Re-stating your aims in a clearer way might also help the reader tie in the findings in the results and discussion

Methods

The data has been analysed appropriately by accounting the for clustered nature of the data, i.e. articles clustered within journals.

3. You may wish to indicate what Models 1-5 were in this section

4. Have you considered other journal information – e.g. open access, print vs online publication, average publication times?

5. Kaplan-Meier nonparametric estimator plots stratified times to acceptance, unadjusted for other factors. Have you considered using Cox proportional hazards regression to plot adjusted times?

6. The benefit of using clustered analysis methods is that you can interpret variation in the outcome by each cluster. Stepwise approach of adding valuables sequentially and looking at the variation in the outcome might be useful to help in explain proportion of variation attributable to journal and author-factors, and then seeing which factor specifically explains the found variation. The paper would benefit from adding interpretation of the random effects in the results section.

7. For gender hypothesis, it would have been interesting to look at the publication rates pre and post COVID by gender, to see if rates for female authors have dropped during COVID times.

8. To test whether there is a differential in the duration of peer-review process for women and men the use of propensity matching techniques would have been more appropriate

9. Have you considered counting the number of authors, or the corresponding author seniority in the models? (noting that the latter may not be able to be ascertained). Single author papers vs multi-author papers would have different responses to peer reviewer comments.

Results

Generally well written and supported by tables/figures.

10. Inclusion of a descriptive average/median times by some key factors at the start of the results session can help the reader understand the data better, prior to interpreting model outputs

11. Inclusions of random effect interpretation would help in describing the variation and explaining which factors drive it. Please see my comment 6.

12. You have excluded a considerable amount of data based on non-assignment of author’s gender. Have you performed analyses on the data without adjusting for gender to see it the institutional disparities exits?

Discussion

13. This section could be strengthened by further elaboration of how the results tie in with the existing literature. Gender related differences hypothesis could have benefited from a different analysis, as well as a discussion around what other factors, beyond those measured in the study, could have contributed to the findings

14. Further discussion about publication rates for covid vs non-covid articles could have been interesting, as well as a discussion about characteristics of COVID published papers. A limitation of the inclusion of 8% (8848/106116) of found COVID papers in the analysis should also be mentioned.

15. Discussion should also acknowledge that peer review process is likely to have been impacted by other factors not able to be measured (e.g. editorial load, editorial processes, editor gender, (using the same argument as in the article), number of reviewers reviewing the article etc).

Reviewer #4: Dear Authors;

The manuscript is in interesting field, and well designed. However, it is essential to consider following comments to improve its quality.

-Whole manuscript is required to English language editing.

-Title should be brief.

-Please Consider re-building and homogeneous addressing Ref. in the manuscript according to Journal Instruction Authors.

-Please describe the details of all 6 models in method section.

-Some sections contain un-neccesary data such as description on single and blind peer-review.

-What is your mention of "unknown gender"? If it is really unknown, I suggest to remove their data from the tables.

-As you know, other factors such as "subject area and type of study"; drug discovery, diagnostic tests, treatment approaches, etc have impact on speed of review, and gender disparities (below papers). It is suggested to consider their impact in your models.

-In appendix, it is not necessary description and Ref.

-Discussion should be re-written, needs to more details, compare to previous studies, overall needs to deep revision. following studies may be useful, in addition, the current study is not the first study, please correct:

1-Scientometric analysis of medical publications during COVID-19 pandemic: the twenty-twenty research boom.

Aggarwal A, Agosti E, Singh PM, Varshini A, Garg K, Chaurasia B, Zanin L, Fontanella MM. Minerva Med. 2021 Oct;112(5):631-640. doi: 10.23736/S0026-4806.21.07489-9.

2-Gender Disparities in Authorship of Invited Manuscripts During the COVID-19 Pandemic.

Brown C, Novick TK, Jacobs EA. Womens Health Rep (New Rochelle). 2021 May 25;2(1):149-153. doi: 10.1089/whr.2021.0023

3- Characteristics of academic publications, preprints, and registered clinical trials on the COVID-19 pandemic.

Gianola S, Jesus TS, Bargeri S, Castellini G. PLoS One. 2020 Oct 6;15(10):e0240123. doi: 10.1371/journal.pone.0240123.

4-Global academic response to COVID-19: Cross-sectional study.

Helliwell JA, Bolton WS, Burke JR, Tiernan JP, Jayne DG, Chapman SJ. Learn Publ. 2020 Jul 1:10.1002/leap.1317. doi: 10.1002/leap.1317.

5-Impact of the Early COVID-19 Pandemic on Gender Participation in Academic Publishing in Radiation Oncology.

Anabaraonye N, Tsai CJ, Saeed H, Chino F, Ekpo E, Ahuja S, Garcia O, Miller RC. Adv Radiat Oncol. 2021 Nov 18;7(2):100845. doi: 10.1016/j.adro.2021.100845.

-Ref. 30 is incomplete.

-Most of the Ref. are addressed in introduction. Please reduce the numbers.

Best Regards,

6. PLOS authors have the option to publish the peer review history of their article (what does this mean?). If published, this will include your full peer review and any attached files.

Reviewer #1: No

Reviewer #2: No

Reviewer #3: **Yes: **Sanja Lujic

Reviewer #4: No

---

## [Author Response · Author response to Decision Letter 0]

22 Aug 2022

Please find the response to the reviewers in the attached files.

---

## [Decision Letter · Decision Letter 1]

20 Sep 2022

PONE-D-22-07162R1Institution and gender-related differences in publication speed before and during COVID-19PLOS ONE

Dear Dr. Holding,

Thank you for submitting your manuscript to PLOS ONE. After careful consideration, we feel that it has merit but does not fully meet PLOS ONE’s publication criteria as it currently stands. Therefore, we invite you to submit a revised version of the manuscript that addresses the points raised during the review process.

We look forward to receiving your revised manuscript.

Kind regards,

Negar Rezaei, M.D., Ph.D.,

Academic Editor

PLOS ONE

Reviewers' comments:

Reviewer's Responses to Questions

**Comments to the Author**

1. If the authors have adequately addressed your comments raised in a previous round of review and you feel that this manuscript is now acceptable for publication, you may indicate that here to bypass the “Comments to the Author” section, enter your conflict of interest statement in the “Confidential to Editor” section, and submit your "Accept" recommendation.

Reviewer #1: All comments have been addressed

Reviewer #3: All comments have been addressed

Reviewer #4: (No Response)

2. Is the manuscript technically sound, and do the data support the conclusions?

Reviewer #1: Yes

Reviewer #3: Yes

Reviewer #4: Partly

3. Has the statistical analysis been performed appropriately and rigorously? 

Reviewer #1: Yes

Reviewer #3: Yes

Reviewer #4: I Don't Know

4. Have the authors made all data underlying the findings in their manuscript fully available?

Reviewer #1: Yes

Reviewer #3: Yes

Reviewer #4: Yes

5. Is the manuscript presented in an intelligible fashion and written in standard English?

Reviewer #1: Yes

Reviewer #3: Yes

Reviewer #4: No

6. Review Comments to the Author

Reviewer #1: (No Response)

Reviewer #3: Thank you for taking the time to address my comments and suggestions in a thoughtful and detailed manner.

I have no more comments to add, aside from minor fixes on page 4:

Introduction, sentence "Using recently available data on the peer-review duration of 78,085 medical papers", it's customary not to present N's in this section

Materials and methods, final sample of COVID-19 papers should be consistent (8848 stated first, then 8828).

Reviewer #4: Dear Authors,

Although most of the comments are considered, there are some concerns as following.

-Needs to language editing. For example, page 5, line 2: "for further specifications on the underlying methodology, see (43)".

-In the last paragraph of the introduction, it is repeated aims. Please rewrite this paragraph and summarize.

-Please clarify, finally, you considered data of "gender unknown", and its results in figures, and interpretation of data or not? If your answer is "yes", it is better to exclude the data, because your main aim is determining the effect of gender on publish speed. If your answer is "NO", that is OK.

-As there are several papers in the field, what is the novelty of your study? In this regard, I suggested compare your results with the others.

Best Regards,

7. PLOS authors have the option to publish the peer review history of their article (what does this mean?). If published, this will include your full peer review and any attached files.

Reviewer #1: No

Reviewer #3: No

Reviewer #4: No

---

## [Author Response · Author response to Decision Letter 1]

4 Oct 2022

Dear Reviewers,

Thanks to all of you for your valuable comments and suggestions. These have clearly improved the paper. Reviewer 1-3 found the previous submission ready for publication. Reviewers 3 and 4 had a few smaller requests for edits that we address below.

Reviewer #3:

I have no more comments to add, aside from minor fixes on page 4:

Introduction, sentence "Using recently available data on the peer-review duration of 78,085 medical papers", it's customary not to present N's in this section

Thank you for this suggestion, we have now removed the sample size from the introduction section.

Materials and methods, final sample of COVID-19 papers should be consistent (8848 stated first, then 8828). 

Thanks for highlighting these inconsistencies in the materials and methods section. The typo has now been corrected and the sample size of COVID-19 papers is now consistent across the paper.

Reviewer #4:

Although most of the comments are considered, there are some concerns as following.

1. Needs to language editing. For example, page 5, line 2: "for further specifications on the underlying methodology, see (43)".

Thank you for this suggestion. We now have updated P4, line 1.

2.In the last paragraph of the introduction, it is repeated aims. Please rewrite this paragraph and summarize.

Thank you for raising this point. We have changed the last paragraph of the introduction by removing the double references to the aim of the study:

P3, line 33: “In this study, we adopt a new perspective on the widening disparities during COVID-19 by using recently available data on the peer-review duration of metical papers. We aim to examine whether the pandemic has widened gender and institution-related differences in the average time from manuscript submission to acceptance. Additionally, given that some peer review processes differ in terms of whether institutional status queues and/or gender are visible to reviewers (due to having either single or double blinded review), we also descriptively investigate whether double blind peer review reduces institution/gender biases in publication speed during the pandemic.”

3.Please clarify, finally, you considered data of "gender unknown", and its results in figures, and interpretation of data or not? If your answer is "yes", it is better to exclude the data, because your main aim is determining the effect of gender on publish speed. If your answer is "NO", that is OK.

We think there may have been some confusion. We are not interested in the analysis of the “gender unknow” category. The paper included this category to avoid excluding from the analysis all the authors for which the three name-to-gender assignment algorithms did not provide reliable results. Indeed, despite for these authors the gender category was “unknown” (hence not discussed as a relevant category in our gender analysis); these authors were still included in the institution-related difference analysis. This allowed to maintain a full sample size in the analysis also for authors whose gender wasn’t captured by the algorithm in the analysis. Therefore, to answer your primary question we are not considering the category “gender unknown” in our analysis. As confirmed by the results and discussion section, no reference to this category is included in the presentation and discussion of the findings. We now include a further clarification about this in the paper P5, line: 18.

4.As there are several papers in the field, what is the novelty of your study? In this regard, I suggested compare your results with the others. 

Thank you for this suggestion. The novelty of the paper lies in analysing the widening disparities during COVID-19 by examining changes in the duration of the peer-review process for women and men, and for scientists at different strata of the institutional prestige hierarchy. Additionally, we also descriptively investigate whether double blind peer review reduces institution/gender biases in publication speed during the pandemic. As stated in the paper: 

P11, line 16: “The current study aimed to assess whether the pandemic had widened existing institution and gender-related differences in publishing speed, by using bibliometric metadata on the duration of peer-review. While prior studies have used such data to document increases in the average publication speed during COVID-19 (Else, 2020; Helliwell et al., 2020; Kun, 2020; Zhang et al., 2020), none have examined how changes in journal turn-around times varied by gender and institution status.”. 

In our previous resubmission, we extended the literature review and discussion to cover the relevant papers suggested by reviewers 1, 3 and 4. Moreover, we included additional studies published after our initial submission. Since this paper is an empirical study (and not a review study), we cannot cover all existing bibliometric papers on COVID-19. This literature is huge and still growing. Instead, we have limited our focus to published work of direct relevance to our research question. We welcome any further suggestions on literature that directly relates to our research question but feel confident that we have covered all central contributions.

---

## [Decision Letter · Decision Letter 2]

19 Oct 2022

Institution and gender-related differences in publication speed before and during COVID-19

PONE-D-22-07162R2

Dear Dr. Holding,

We’re pleased to inform you that your manuscript has been judged scientifically suitable for publication and will be formally accepted for publication once it meets all outstanding technical requirements.

Kind regards,

Negar Rezaei, M.D., Ph.D.,

Academic Editor

PLOS ONE

Additional Editor Comments (optional):

Dear Authors,

Please edit line 33-34 page 4: "and has been shown to be ~96% accurate (for further specification on the underlying methodology, see (43))." to "and has been shown to be ~96% accurate (43)."

Best Regards,

Reviewers' comments:

Reviewer's Responses to Questions

**Comments to the Author**

1. If the authors have adequately addressed your comments raised in a previous round of review and you feel that this manuscript is now acceptable for publication, you may indicate that here to bypass the “Comments to the Author” section, enter your conflict of interest statement in the “Confidential to Editor” section, and submit your "Accept" recommendation.

Reviewer #4: All comments have been addressed

2. Is the manuscript technically sound, and do the data support the conclusions?

Reviewer #4: Yes

3. Has the statistical analysis been performed appropriately and rigorously? 

Reviewer #4: Yes

4. Have the authors made all data underlying the findings in their manuscript fully available?

Reviewer #4: Yes

5. Is the manuscript presented in an intelligible fashion and written in standard English?

Reviewer #4: Yes

6. Review Comments to the Author

Reviewer #4: Dear Authors,

Please edit line 33-34 page 4: "and has been shown to be ~96% accurate (for further specification on the underlying methodology, see (43))." to "and has been shown to be ~96% accurate (43)."

Best Regards,

7. PLOS authors have the option to publish the peer review history of their article (what does this mean?). If published, this will include your full peer review and any attached files.

Reviewer #4: No

---

## [Editor Report · Acceptance letter]

28 Oct 2022

PONE-D-22-07162R2 

Institution and gender-related differences in publication speed before and during COVID-19 

Dear Dr. Holding:

I'm pleased to inform you that your manuscript has been deemed suitable for publication in PLOS ONE. Congratulations! Your manuscript is now with our production department. 

Kind regards, 

on behalf of

Dr. Negar Rezaei 

Academic Editor

PLOS ONE